# Significant adhesion reduction and time saving in pediatric heart surgery with 4DryField PH: A retrospective, controlled study

Robert Cesnjevar[1], Ariawan Purbojo[2], Claas Haake[3], Joachim Laas[3,4]*

1 Department for Heart Surgery, University Children's Hospital Zurich, Zurich, Switzerland, 2 Department for Pediatric Heart Surgery, University Clinic Erlangen, Erlangen, Germany, 3 PlantTec Medical GmbH, Lüneburg, Germany, 4 Centre Surgery, Hanover Medical School, Hanover, Germany

* Joachim.laas@planttec-medical.de

**Data Availability Statement:** All relevant data are within the paper and its Supporting information files.

## Abstract

Adhesions formation after surgery for congenital heart defects can complicate follow-up procedures due to bleeding from detached adhesion bands, injury to cardiac structures or large vessels, all of which do prolong operation times. The problem is enhanced by the fact that detached adhesions are predilection sites for new adhesions setting off a downward spiral. 4DryField® PH gel barrier has demonstrated high efficacy in reducing postoperative adhesions in general surgical and gynecological studies. This retrospective controlled study of 22 patients evaluates whether these positive results can be confirmed in pediatric cardiac surgery. Adhesions were scored from photographs of follow-up interventions by an independent cardiac surgeon blinded to group assignment. The publication provides not only score numbers but also original photographs of all sites for better traceability and transparency. In addition, timesaving due to reduced adhesions was evaluated. Results show a significantly reduced adhesion score for the 4DryField® group. Importantly, this resulted in a significantly shorter period between skin incision and start of cardiopulmonary bypass. In addition, time-saving due reduced adhesion formation was evaluated. The use of 4DryField® was safe, although higher doses per kg were used than in adults.

## Introduction

The prevalence of congenital heart defects (CHD) in the European Union is estimated to be 36,000 per year [1]. Those defects commonly require surgical interventions, in which one of the postoperative long-term problems is the formation of adhesions. Histological and ultra-structural studies show that surgical trauma and fibrin bands of hematomas serve as scaffold for the collagenous adhesion bands [2, 3]. Although the pericardial mesothelium has fibrino-lytic activity that can dissolve fibrinous adhesions, this mechanism is not always sufficient to prevent adhesions and agglutinations.

**Funding:** The authors received no specific funding for this work.

**Competing interests:** I have read the journal's policy and the authors of this manuscript have the following competing interests: Claas Haake is a paid employee of PlantTec Medical GmbH. Joachim Laas is a paid employee and board member of PlantTec Medical GmbH. This does not alter our adherence to PLOS ONE policies on sharing data and materials. PlantTec Medical GmbH is the manufacturer of the adhesion barrier 4DryField PH, which has been investigated in this study.

In addition, it has been shown that surgical trauma has an inhibitory effect on fibrinolytic activities [4]. Hence, early fibrinous adhesions are infiltrated by proliferating fibroblasts. Histologically, in their full formation adhesion bands may contain fibrocytes, collagen fibers and blood vessels [5, 6]. An additional impact factor is the usage of cardiopulmonary bypass (CPB) techniques since extracorporeal circulation may impair fibrinolytic activity of the native pericardium. In addition, epicardial irritation and petechiae can cause severe tissue reactions and trigger adhesions [7–9]. This is particularly problematic when repeated operations are necessary, such as in children with hypoplastic left heart syndrome. The problem is exacerbated by the fact that the adhesions once developed have to be resolved in subsequent surgeries and that the detachment sites themselves are predilection sites for adhesions. The problems caused by the presence of adhesions are not only limited to its hazardous effect on the procedure of re-sternotomy, but there is also evidence of adhesions causing ventricular dysfunction [10] by compromised right ventricular contraction triggered by adhesions between the heart and the underside of the sternum. These can also lead to restricted left ventricular diastolic filling [10, 11]. A high risk factor for the development of tight adhesions to the sternum is the non-closure of the pericardium [7]. In particular, the right ventricle, the right atrium, the aorta and the innominate vein can adhere to the sternum and are, thus, endangered during re-opening [6].

Considering these problems, a device that can reliably prevent adhesion formation after pericardial surgery, especially in pediatric heart surgery, would be desirable. This publication presents the results of the application of the medical device 4DryField® PH (PlantTec Medical GmbH, Lüneburg, Germany) in this area. 4DryField® PH consists of hydrophilic microparticles, which have to be transformed into a gel by dripping with saline solution before application for adhesion prevention. This gel then forms a mechanical barrier that separates the wound areas and, thus, facilitates the healing of the mesothelial lining of the pericardium. The adhesion prevention properties of 4DryField® PH been demonstrated in animal models [12–18], as well as through its successful clinical application as an adhesion prevention agent in general surgical and gynecological procedures [19–26].

## Methods and materials

This retrospective, controlled, observational study includes 22 patients who underwent surgery between 2013 and 2016. Eleven of these patients (patients 1–11) received the barrier gel 4Dry-Field® PH for adhesion prevention under standard of care and the other eleven (patients A-K) did not receive an adhesion prevention device. Only patients with multi-stage procedures that allowed direct assessment of adhesion formation during subsequent surgery were included in the study. A summary of the respective procedures for all patients can be found in Table 1. The 22 patients included were operated by two different surgeons. Patients in the intervention group received 5 g of 4DryField® PH powder per surgery, which was transformed into a gel in-situ by dripping with sterile 0.9% sodium chloride solution. It was ensured that all accessible adhesion predilection sites were covered. No other adhesion prevention measures (like the usage of GoreTex® membranes) were applied in any of the patients.

During most of the surgeries, photographic documentation of the adhesion predilection sites and the respective sites during re-operation was performed so that post-operative adhesion scoring could be performed by pre-post comparisons. Patients who did not have photographic documentation of the re-operation were not included in the analysis of the effectiveness of adhesion prevention. An independent cardiac surgeon blinded to the decision of use of 4DryField® adhesion prevention device performed the adhesion scoring. A five-point score ranging from 1 to 5 and including half point values was used similar to Konertz et al. [27] and Walther et al. [28] (1: no adhesions, 2: filmy adhesions, 3: moderate adhesions,

**Table 1. Surgical procedures and corresponding intermediate time intervals.**

| Patient | First surgery | t | Second surgery | t | Third surgery |
|---|---|---|---|---|---|
| 1 | RVOT muscle resection and enlargement plasty | 0.5 | Closure of ventricular septal defect | | |
| 2 | Partial correction of persistent truncus arteriosus | 8.5 | Correction of persistent truncus arteriosus | | |
| 3 | Closure of ventricular septal defect | 8.5 | Slide tracheoplasty, left pulmonary artery reduction surgery, right ventricle to pulmonary artery conduit | | |
| 4 | Aortopulmonary shunt | 4 | Anomalous pulmonary venous connection correction, atrial septectomy, bidirectional Glenn procedure | 2 | Suture-less repair of pulmonary venous obstruction |
| 5 | Norwood I | 6.5 | Shunt exchange | 0.5 | Decortication |
| 6 | Correction of aortic arch, Arteria Lusoria and Left Superior Vena Cava transplanting, patch closure of atrial septal defect II, pulmonary artery banding | 5 | Closure of ventricular septal defect, pulmonary artery debanding, pulmonary artery enhancement surgery | | |
| 7 | Norwood I | 5 | Glenn procedure | | |
| 8 | Norwood I | 3.5 | Glenn procedure, Left pulmonary artery plasty | | |
| 9 | Bilateral banding of pulmonary arteries ("Gießen procedure") | 0 | Biventricular correction of borderline left ventricle | | |
| 10 | Bilateral banding of pulmonary arteries ("Gießen procedure") | 3.5 | Comprehensive stage II procedure, enhancement of aortic arch | | |
| 11 | Aortopulmonary shunt | 4.5 | Bidirectional Glenn procedure, atrial septectomy | | |
| A | Pulmonary artery plasty, aortopulmonary shunt | 5 | Glenn procedure, left pulmonary artery stenosis patch enlargement (matrix patch) | | |
| B | Norwood I with Sano shunt | 4 | Glenn procedure, pulmonary artery plasty | | |
| C | Bilateral pulmonary artery banding | 5 | DKS Anastomosis | | |
| D | Glenn procedure, pulmonary artery plasty | 1 | Left pulmonary artery thrombectomy, Sano shunt | | |
| E | Aortopulmonary shunt | 4.5 | Glenn procedure | | |
| F | Aortopulmonary shunt | 5.5 | Bilateral bidirectional Glenn procedure | | |
| G | Norwood I with Sano shunt | | Bilateral Glenn procedure, pulmonary artery plasty | | |
| H | Aortopulmonary shunt, atrioseptectomy | 4 | Bilateral Glenn procedure, pulmonary artery bifurcation plasty | | |
| I | Aortopulmonary shunt | 5.5 | Bilateral Glenn procedure, atrioseptectomy | | |
| J | Intraluminal banding of pulmonary artery | 5.5 | Bilateral Glenn procedure | | |
| K | Atrioseptectomy, bilateral banding of pulmonary artery | 1 | Norwood I | | |

t = time between the two respective procedures in months, patients 1–11: 4DryField group, patients A–K: control group.

4: dense adhesions, 5: very dense adhesions, necessitating electro-coagulation. If severity differed throughout the evaluated area, respective interjacent scores were given.).

Adhesions were scored at the second and, if applicable, third intervention. Adhesion scores are available for nine interventions of each group. In addition, surgery durations for the re-operations were compared. The duration of the entire surgery, as well as the part from first incision to Cardiopulmonary Bypass (CPB), during which adhesion detachment is performed, were individually evaluated to detect correlations with the adhesion prevention measures.

Data collected for comparison of the two study limbs included sex, age at first and second surgery, BMI, ASA, RACHS and Aristotle scores.

Data evaluation was performed using Microsoft Excel 2016 (Microsoft Corporation, Redmond, Washington, USA) and GraphPad Prism 8 (GraphPad Software Inc., San Diego, California, USA). For the ASA, RACH, Aristotle and adhesion scores, as well as the BMIs and age, the arithmetic mean was first calculated and then determined, whether the respective data

were distributed normally with the D'Agostino-Pearson omnibus normality test. If the data were normally distributed, p-values were calculated using a two-sided unpaired t-test, otherwise with a two-sided Mann-Whitney test. For patients who underwent a third surgery, the results of this third surgery were included in the study as a separate additional entry regarding their adhesion scores (patients in the intervention group received adhesion prevention treatment during each surgery). Sex distribution in the groups was analyzed with Fisher's exact test and surgery durations using the Mantel-Cox test. All results were considered statistically significant when the calculated p-value was below 0.05.

This retrospective study was approved by the Ethics Committee of the Friedrich-Alexander University Erlangen-Nürnberg with the Number 41_18 Bc. The participant's legal guardians gave written informed consent for inclusion in the observational study.

## Results

None of the basic parameter age, sex, BMI, ASA, RACHS or Aristotle score differed significantly between the two study groups. These results are summarized in Table 2.

The mean adhesion score in the control group was 2.6, which shows that the adhesion development after such interventions is usually of medium severity. The use of 4DryField® PH reduced adhesion formation to a score of 1.7. Statistical comparison showed that this difference is significant with a p-value of 0.003. Accordingly, the duration of the first part of the follow up surgery, from the first incision to CPB, which includes the detachment of adhesions, was significantly reduced in the 4DryField® group: from an average of 90 min for the untreated patients to 69 min for the 4DryField® patients. The mean total duration of surgery was not significantly different between the groups. Results for adhesion scores and surgery lengths are shown in Table 3 and representative images of the adhesion development in the two groups are shown in Figs 1 and 2. The image set of S1 Fig shows that the hearts treated with 4DryField® were either almost free of adhesions or showed only filmy adhesions while the images in S2 Fig show that in 7/9 cases, the non-treated pediatric hearts showed adhesions more severe than filmy. A detailed description of the adhesions found is included in the S1 Table.

None of the patients developed any complications attributable to 4DryField® PH usage. In the control group, one patient suffered from endocarditis after the first surgery.

**Table 2. Comparison of the basic parameters of the study groups.**

|  | 4DryField group | | Control group | | |
|---|---|---|---|---|---|
|  | Mean | SD | Mean | SD | p |
| Age (first surgery) [d] | 27 | 35.5 | 44 | 52.5 | 0.999 |
| Age (second surgery) [d] | 163 | 68.6 | 162 | 62.2 | 0.967 |
| BMI [kg/m$^2$] | 13.1 | 1.4 | 12.9 | 3.1 | 0.681 |
| ASA | 3.2 | 0.4 | 3.2 | 0.4 | 0.999 |
| RACHS | 4 | 1.5 | 2.9 | 1.1 | 0.077 |
| Aristotle | 9.2 | 3.3 | 7.5 | 2.4 | 0.311 |
|  | Quantity | % | Quantity | % |  |
| Male sex | 9 | 82 | 6 | 55 | 0.362 |

SD: standard deviation.

**Table 3. Results for the adhesion development and surgery durations.**

|  | 4DryField group | | Control group | | |
| --- | --- | --- | --- | --- | --- |
|  | **Mean** | **SD** | **Mean** | **SD** | **p** |
| Adhesion score | 1.7 | 0.4 | 2.6 | 0.5 | 0.003* |
| Length (first incision to CPB) [min] | 69 | 21 | 90 | 22 | 0.037* |
| Length (whole surgery) [min] | 377 | 127 | 357 | 78 | 0.448 |

SD: standard deviation,

*: statistically significant.

## Discussion

In staged pediatric cardiac surgery, adhesions in the pericardium make transsternal reopening of the chest and exposure of the heart and great vessels difficult. The present study was able to demonstrate that adhesion formation was visually significantly reduced after using the gel barrier 4DryField® PH, also confirmed by score evaluation from a blinded cardiac surgeon. The study's expressiveness is limited though by its small patient number and single-center design.

Scores to determine the extent of adhesions are an established technique to describe adhesion formation. Comparison of such scores can be used to statistically evaluate the efficacy of medical devices. The assessment of adhesions is always a subjective process by the assessor, and by its nature, as with any abstraction, information is always lost.

The present work is accompanied by the images of all patients used for evaluation. On the one hand, this is intended to fill the information gap created by the subjective assessment. On the other hand, in the previously published works on adhesion prophylaxis in pediatric cardiac

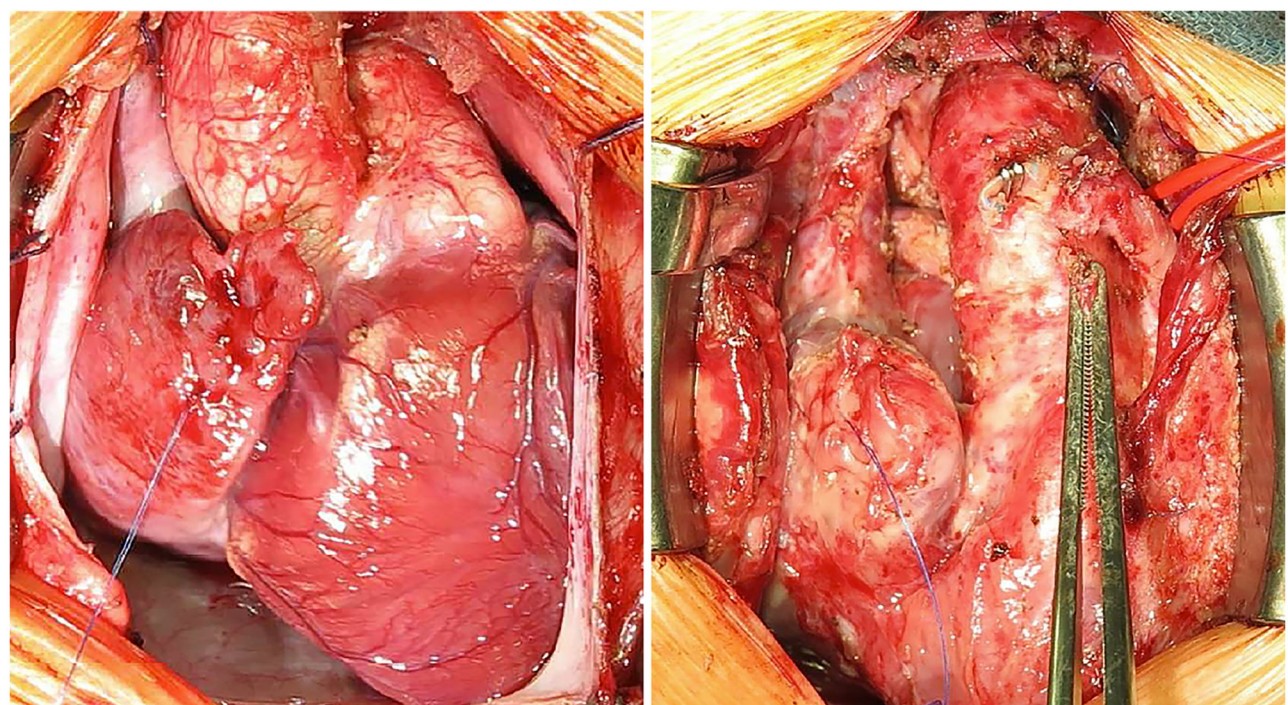

**Fig 1. Representative photographs from the first (left) and second (right) surgery of the same patient from the control group.**

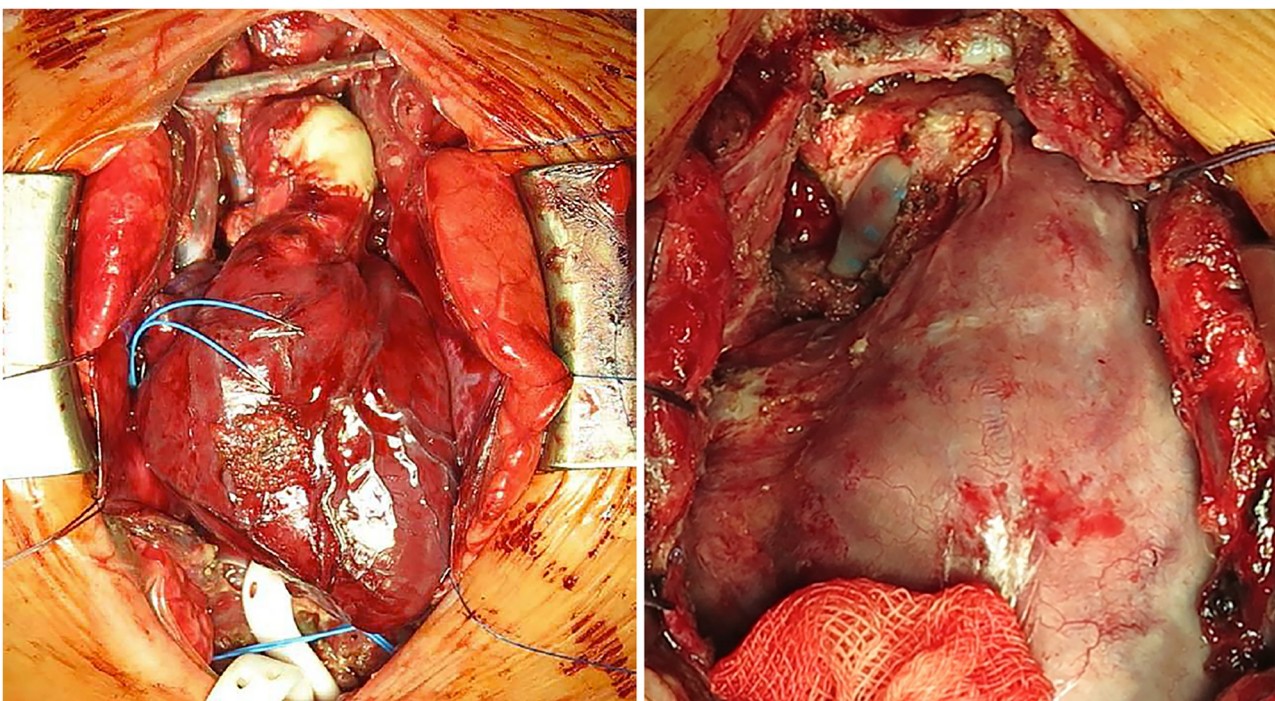

**Fig 2. Representative photographs from the first (left) and second (right) surgery of the same patient from the 4DryField group.**

surgery, there is no adequate image material so far, as it is the case for example in gynecology. The images are to make the score evaluation comprehensible.

The reduced adhesions have a positive effect on the surgical procedure, as evidenced by the significant timesaving in the initial preparation phase prior to starting extracorporeal circulation. It must be noted though, that the surgeons performing the reinterventions were not blinded to group allocation representing another limitation in the study design. Furthermore, the fact that two different surgeons performed the study surgeries possibly influenced the usage of electro-coagulation and, thereby, adhesion scoring. In addition to timesaving, effective prevention of adhesion formation decreases the risk of injury to the heart and great vessels. The indifference of total operation time between the two groups is explained by the wide range of surgical indications leading to a high variance in surgery times independent of the necessity of adhesiolysis. A larger, randomized study with narrow inclusion criteria will likely achieve statistical significance here as well.

Studies on adhesion prophylaxis with other medical devices in pediatric cardiac surgery have been published on Seprafilm®, Coseal®, and REPEL-CV®.

A retrospective study published in 2005 reported the use of Seprafilm® for adhesion prophylaxis in 350 children undergoing pediatric cardiac surgery. Thirty of these children were evaluated at reoperation with a subjective tenacity and extent scoring system, which were then compared to 10 random children not treated. No significant difference was calculated concerning the adhesion scores. The study showed a significantly shortened operative time from skin incision to the start of extracorporeal circulation [28]. Our study shows both a significant difference in adhesion formation and a significantly shortened operation time.

In a study published in 2009 using CoSeal® Surgical Sealant (Baxter, Deerfield, Illinois, USA) for adhesion prophylaxis, 36 children underwent adhesion evaluation at follow-up surgery. A mean adhesion score of 8.3 on a scale of 0 to 21 was determined [29]. The score of 1.7

on a scale of 1 to 5 obtained in the present study appears more favorable. The study with Coseal® lacks a control group, limiting comparison. In addition, this study recorded five serious adverse events, including two pericardial tamponades, that were attributed to CoSeal® usage.

In a multicenter randomized evaluator-masked trial of the by now discontinued REPEL-CV® membrane (SyntheMed Inc., Iselin, New Jersey, USA) in 103 neonates with planned resternotomies, the second look showed a mean score of 1.9 on a scale from 0 to 3 [30]. The results with 4DryField® appear superior, but the small patient numbers limit comparability.

The literature comparison shows a very positive evaluation of the effectiveness of the gel barrier 4DryField® for the prophylaxis of pericardial adhesions in pediatric cardiac surgery. Equally positive is the tolerability and safety of 4DryField®. Each patient, regardless of body weight, had received 5 g of product, which corresponds to a dose of up to 2 g per kg body weight and thus a much higher dose than ever reported for adults [19–25, 31–33]. Complications did not occur. No product residues were found during follow-up operations.

## Conclusion

The use of 4DryField® as a gel barrier leads to a significant reduction in adhesion formation, resulting in a significant reduction in preparation time in the pre-bypass period during re-interventions. Its tolerability and safety can be considered problem-free. Further prospective controlled studies are desirable to confirm the promising results.

## Supporting information

**S1 Fig. Representative photographs of all patients scored for adhesion development in the 4DryField group.** Hover over subfigures for patient and surgery attribution.
(PDF)

**S2 Fig. Representative photographs of all patients scored for adhesion development in the control group.** Hover over subfigures for patient and surgery attribution.
(PDF)

**S1 Table. Attributed adhesion scores with description for all patients scored for adhesion development in both groups.**
(PDF)

## Author Contributions

**Conceptualization:** Robert Cesnjevar, Joachim Laas.

**Data curation:** Ariawan Purbojo, Claas Haake.

**Funding acquisition:** Joachim Laas.

**Investigation:** Robert Cesnjevar, Ariawan Purbojo.

**Methodology:** Robert Cesnjevar.

**Project administration:** Robert Cesnjevar, Joachim Laas.

**Resources:** Robert Cesnjevar, Joachim Laas.

**Supervision:** Robert Cesnjevar.

**Validation:** Robert Cesnjevar.

**Writing – original draft:** Robert Cesnjevar, Ariawan Purbojo.

**Writing – review & editing:** Robert Cesnjevar, Ariawan Purbojo, Claas Haake, Joachim Laas.

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
