## [Decision Letter · Decision Letter 0]

29 Jun 2022

PONE-D-21-34247Significant adhesion reduction and time saving in pediatric heart surgery with 4DryField^®^ PH: a retrospective, controlled studyPLOS ONE

Dear Dr. Laas,

Thank you for submitting your manuscript to PLOS ONE. After careful consideration, we feel that it has merit but does not fully meet PLOS ONE’s publication criteria as it currently stands. Therefore, we invite you to submit a revised version of the manuscript that addresses the points raised during the review process.

We look forward to receiving your revised manuscript.

Kind regards,

Vanessa Carels

Staff Editor

PLOS ONE

Journal Requirements:

I have read the journal's policy and the authors of this manuscript have the following competing interests:

Claas Haake is a paid employee of PlantTec Medical GmbH.

Joachim Laas is a paid employee and board member of PlantTec Medical GmbH.

PlantTec Medical GmbH is the manufacturer of the adhesion barrier 4DryField PH, which has been investigated in this study.

Reviewers' comments:

Reviewer's Responses to Questions

**Comments to the Author**

1. Is the manuscript technically sound, and do the data support the conclusions?

Reviewer #1: Yes

2. Has the statistical analysis been performed appropriately and rigorously? 

Reviewer #1: Yes

3. Have the authors made all data underlying the findings in their manuscript fully available?

Reviewer #1: Yes

4. Is the manuscript presented in an intelligible fashion and written in standard English?

Reviewer #1: Yes

5. Review Comments to the Author

Reviewer #1: Cesnjevar et al submitted a single center retrospective cohort study titled "Significant adhesion reduction and time saving in pediatric heart surgery with 4DryField PH: a retrospective, controlled study". Postoperative adhesion formation is a major problem in pediatric cardiac surgery which leads to a) increased operative times b) increased risk of bleeding and critical structure injuries. In this single center study, authors used 4DryField PH gel (prepared by mixing 5 grams of powder in normal saline solution) and applying the gel at the end of the first surgery. At the reoperation, a blinded surgeon evaluated and scored the adhesion formation. In addition, authors also used the photographic evidence to assess adhesion formation. In this study, total of 22 children were evaluated, 50% (n=11) received 4DryField PH gel and the rest (n=11) served as a controlled group. A significant reduction in adhesion formation was noted (based on the scoring) and improved skin incision to cardiopulmonary bypass times were also noted in 4H group. Below are my thoughts/ suggestions:

1. Overall it is a well done study, specially including the photographs helps the reader a lot.

2. Regarding the statistically significant reduction in time from skin incision to CPB, there is a confounding variable as the operator was not blinded. Consider highlighting that or include that in the limitation section.

3. Consider having indicators with the pictures

4. Small number and single center study should also be included in the limitation section.

Thank you.

6. PLOS authors have the option to publish the peer review history of their article (what does this mean?). If published, this will include your full peer review and any attached files.

Reviewer #1: No

---

## [Author Response · Author response to Decision Letter 0]

8 Jul 2022

Dear Editor and Reviewer,

thank you very much for your helpful comments. 

Concerning the points raised regarding journal requirements and our corresponding actions:

We ensured that our manuscript meets PLOS ONE's style requirements by adjusting the names of the figure files, the way tables are cited, the citation style, the format of Supporting Information Citations and the affiliations of the first and fourth author.

2. Thank you for stating the following in the Competing Interests section.

We revised the Competing Interests statement as suggested. You can find the full statement in the cover letter.

Concerning the points raised by the reviewer under point 5 “Review Comments to the Author” and our corresponding actions:

• Regarding the statistically significant reduction in time from skin incision to CPB, there is a confounding variable as the operator was not blinded. Consider highlighting that or include that in the limitation section.

We added a sentence to the discussion section starting at line 86 indicating that the fact that the surgeon was not blinded is a confounding variable concerning the result of a statistically significant reduction in time from skin incision to CPB and, therefore, a limitation.

• Consider having indicators with the pictures

We did not add indicators to the pictures. Indicators have been omitted from the illustrations in the publication to keep as many structures visible as possible. Unless absolutely necessary, we would therefore like to keep the illustrations as they are.

• Small number and single center study should also be included in the limitation section.

We added a sentence in the discussion section starting at line 71 pointing out the small patient number and single center design as limitations.

We hope that the manuscript is now acceptable for publication in PLOS ONE.

Yours Sincerely

Prof. Dr. med. Joachim Laas

---

## [Decision Letter · Decision Letter 1]

2 Sep 2022

PONE-D-21-34247R1Significant adhesion reduction and time saving in pediatric heart surgery with 4DryField^®^ PH: a retrospective, controlled studyPLOS ONE

Dear Dr. Laas,

Thank you for submitting your manuscript to PLOS ONE. After careful consideration, we feel that it has merit but does not fully meet PLOS ONE’s publication criteria as it currently stands. Therefore, we invite you to submit a revised version of the manuscript that addresses the points raised during the review process.

We look forward to receiving your revised manuscript.

Kind regards,

Aditya Badheka

Guest Editor

PLOS ONE

Journal Requirements:

Reviewers' comments:

Reviewer's Responses to Questions

**Comments to the Author**

1. If the authors have adequately addressed your comments raised in a previous round of review and you feel that this manuscript is now acceptable for publication, you may indicate that here to bypass the “Comments to the Author” section, enter your conflict of interest statement in the “Confidential to Editor” section, and submit your "Accept" recommendation.

Reviewer #2: All comments have been addressed

2. Is the manuscript technically sound, and do the data support the conclusions?

Reviewer #2: Yes

3. Has the statistical analysis been performed appropriately and rigorously? 

Reviewer #2: Yes

4. Have the authors made all data underlying the findings in their manuscript fully available?

Reviewer #2: Yes

5. Is the manuscript presented in an intelligible fashion and written in standard English?

Reviewer #2: Yes

6. Review Comments to the Author

Reviewer #2: Adhesions are difficult to measure and interpret based on subjective nature, but the authors have done a good job of making a scientific objective criteria to grade the adhesions (blinded) then support the findings with operative times.

Although the lack of blinding to prior surgery is a significant confounder as noted in a previous review, it is noted in the paper and there is not a good way to avoid with a single institution study. I also think the heterogeneity of cases can change adhesion formation, especially around conduits and shunts, although since the operations are described well this can at least be accounted for.

Specific comments:

1) I assume no gore membranes were used, even over the rv-pa homografts? These can affect adhesion formation.

2) Isn't the definition of score 5 adhesion (electro coagulation) based on surgeon preferences rather than visualization? It may be worth noting how many different surgeons were participating in the study to frame the subjective factor.

3) Line 119 you should clarify is the adhesive barrier was reapplied after each subsequent surgery.

4) Line 142 why was the mean surgery duration not different? Seems like a statistical change since the time was 20 minutes shorter, but relative to the overall duration would need larger n to show significance.

5) Line 212 grammer can be improved,such as a "shows a" instead of "comes to a"

6) Since the authors have a COI involvement with the product, are they also the ones doing the review and unblinded operations focusing on pre-bypass times? If so this could skew the results. If the authors were not the surgeons on these cases, this should be stated to alleviate any COI concerns.

7. PLOS authors have the option to publish the peer review history of their article (what does this mean?). If published, this will include your full peer review and any attached files.

Reviewer #2: No

---

## [Author Response · Author response to Decision Letter 1]

23 Sep 2022

Dear Editor and Reviewer,

thank you very much for your helpful comments. 

Concerning the points raised by the reviewer under point 6 “Review Comments to the Author” and our corresponding actions:

1) I assume no gore membranes were used, even over the rv-pa homografts? These can affect adhesion formation.

The assumption is correct. We clarified this by adding a sentence to the Methods and materials chapter stating that no other adhesion prevention measures like the usage of GoreTex membranes were used in any of the patients.

2) Isn't the definition of score 5 adhesion (electro coagulation) based on surgeon preferences rather than visualization? It may be worth noting how many different surgeons were participating in the study to frame the subjective factor.

We added a sentence to the Methods and materials section stating that two different surgeons operated the patients included in the study. Furthermore, a statement was added to the discussion section listing this as a further limitation. 

3) Line 119 you should clarify is the adhesive barrier was reapplied after each subsequent surgery.

A clarification was added stating that patients in the intervention group received adhesion prevention treatment during each surgery.

4) Line 142 why was the mean surgery duration not different? Seems like a statistical change since the time was 20 minutes shorter, but relative to the overall duration would need larger n to show significance.

Although the overall surgery duration was 20 minutes shorter, statistical examination performed as described in the manuscript did not yield a significant difference. This is due to the high variance of surgery length which in turn is due to different kind of procedures being included while the patient number is small. The respective sentence in the discussion section has been supplemented with further explanation as well as the outlook that a larger study should be able to receive statistical significance here as well.

5) Line 212 grammer can be improved, such as a "shows a" instead of "comes to a"

The recommendation has been implemented. 

6) Since the authors have a COI involvement with the product, are they also the ones doing the review and unblinded operations focusing on pre-bypass times? If so this could skew the results. If the authors were not the surgeons on these cases, this should be stated to alleviate any COI concerns.

Authors with COI involvement (authors 3 and 4) have neither been involved in adhesion scoring nor in any of the surgeries. A respective declaration has been added to the competing interest statement. Furthermore, the Methods and Materials section now states that adhesion scoring was performed by an independent cardiac surgeon. 

We hope that the manuscript is now acceptable for publication in PLOS ONE.

Yours Sincerely

Prof. Dr. med. Joachim Laas

---

## [Editor Report · Decision Letter 2]

31 Oct 2022

Significant adhesion reduction and time saving in pediatric heart surgery with 4DryField^®^ PH: a retrospective, controlled study

PONE-D-21-34247R2

Dear Dr. Joachim Laas,

We’re pleased to inform you that your manuscript has been judged scientifically suitable for publication and will be formally accepted for publication once it meets all outstanding technical requirements.

Kind regards,

Aditya Badheka

Guest Editor

PLOS ONE

---

## [Editor Report · Acceptance letter]

8 Nov 2022

PONE-D-21-34247R2 

Significant adhesion reduction and time saving in pediatric heart surgery with 4DryField PH: a retrospective, controlled study 

Dear Dr. Laas:

I'm pleased to inform you that your manuscript has been deemed suitable for publication in PLOS ONE. Congratulations! Your manuscript is now with our production department. 

Kind regards, 

on behalf of

Dr. Aditya Badheka 

Guest Editor

PLOS ONE